## Research Article

anxiety; depression; schools; group intervention; child mental health

**Corresponding author:**
Sherinah Saasa;
Email: sherinah_saasa@byu.edu

# Efficacy of a school-based mental health intervention among Zambian youth: a cluster-randomized controlled trial

Sherinah Saasa[1] , Kaitlin P. Ward[2], Cleopas G. Sambo[3], Paula Barrett[4] and Cheuk Yan Lau[5]

[1]School of Social Work, Brigham Young University, Provo, UT, USA; [2]Department of Psychology, School of Social Work, University of Michigan-Ann Arbor, Ann Arbor, MI, USA; [3]Department of Social Work & Sociology, University of Zambia, Lusaka, Zambia; [4]Friends Resilience Programs, Brisbane, Australia and [5]School of Arts and Humanities, Edith Cowan University, Joondalup, Australia

## Abstract

While many children in Africa face notable psychological problems, the majority do not receive needed mental health services. The My FRIENDS Youth Program, a universal cognitive-behavioral intervention for anxiety prevention and resilience enhancement, has demonstrated effectiveness across cultures in children and adolescents. This study explores whether the program's effectiveness extends to Zambian children. Participants were 75 children and adolescents (53% female, ages 10–15) attending low-income schools in Zambia. Four schools were randomly assigned to an intervention ($n = 44$) or waitlist control ($n = 31$). The intervention consisted of 10 weekly sessions plus two booster sessions administered in group format. Assessments were conducted at pre-intervention, immediately post-intervention and 3-month follow-up. Data were analyzed using longitudinal multilevel modeling and controlled for child and parent sociodemographic characteristics. Intervention participation did not lead to reductions in anxiety, depression or parent-child relationship conflict but was associated with reductions in parent-reported internalizing and externalizing symptoms, attention problems and increases in positive parent-child relationships. However, both the intervention and control groups exhibited lower anxiety symptoms from Post-Intervention to 3-Month Follow-Up, suggesting potentially delayed effects. Future research may need to adapt this intervention to meet the needs of children in Zambia.

## Impact statement

We found that participation in the FRIENDS resilience program may be beneficial in reducing mental health symptoms among Zambian children and adolescents. Further involvement in program development and cultural adaptation by local mental health professionals could potentially yield more promising results. This study suggests that school-based interventions to promote the psychological well-being of youth in under-resourced communities are both achievable and well-received by participants.

## Introduction

Mental health problems are among the leading causes of death and health-related disability in children worldwide (Erskine et al., 2015). Despite global attention toward mental health and efforts made by various nations and the World Health Organization (WHO) to address the issue, Africa, home to one of the world's fastest-growing and youngest populations, has largely been overlooked. Evidence indicates that due to compounding factors, children from impoverished backgrounds have a greater risk of developing mental health problems (Coetzee et al., 2022). In sub-Saharan Africa, one in every seven children and adolescents (14.3%) faces significant psychological challenges and one in every ten (9.5%) qualifies for a psychiatric diagnosis (Cortina et al., 2012). A recent systematic review among sub-Saharan African adolescents indicated that approximately 21% had suicidal ideation, anxiety disorders (30%), depression (27%), PTSD (21%) and emotional and behavioral problems (41%) (Jörns-Presentati et al., 2021). Alongside the inherent vulnerability to emotional distress resulting from swift changes in physical and social aspects typical of this developmental stage, the elevated burden of adolescent mental health distress in sub-Saharan Africa is influenced by heightened psychosocial stressors. These stressors encompass persistent exposure to chronic poverty, instances of abuse, encounters with violence and a higher prevalence of specific conditions, such as HIV/AIDS, within the region (Jörns-Presentati et al., 2021; Nyundo et al., 2020).

Despite this, mental health services in sub-Saharan African countries are scarce and the majority of children do not receive the help they need (Atilola et al., 2017; Coetzee et al., 2022; UNICEF, 2021). For example, as of 2020, Zambia had only 10 psychiatrists, 15 psychologists and 425 mental health nurses, amounting to 760 mental health professionals for an entire country population estimated at 19.6 million (The World Bank, 2024; World Health Organization [WHO], 2020). None of these professionals specialized in child or adolescent mental health services (WHO, 2020). In addition, Zambia has one psychiatric hospital and no graduate training for psychologists, occupational therapists and clinical social workers; these numbers have not changed significantly to date (Munakampe, 2020; WHO, 2020). Zambia is experiencing a large demographic shift and is one of the world's youngest countries by median age, with 48% of the population below the age of 15 years (The World Bank, 2024; ZSA, 2018). Its population, much of it urban, has a rapid growth rate of 2.7% per year, reflecting the relatively high fertility rate. As the large youth population attains reproductive age, the population is anticipated to double in the next 25 years, resulting in additional pressure on the demand for jobs, health care and other social services (The World Bank, 2024). Additionally, the Living Conditions Monitoring Survey of 2022 reveals that 60% of the population lives in poverty, which is defined as being unable to meet minimum basic needs consisting of food and essential non-food items within their total income (ZSA, 2022b). Therefore, promoting children's mental health is vital for the nation's future economic growth and social development.

Children's mental health is known to have a crucial interactive relationship with physical health, education, interpersonal relationships and overall healthy development, with long-term implications into adulthood (Centers for Disease Control & Prevention [CDC], 2023). The potential consequences of untreated mental health distress and a lack of education span across domains of reproductive and sexual health, unemployment, homelessness, poverty, adult underachievement and poor quality of life (Best et al., 2006; Birdthistle et al., 2009; Coetzee et al., 2022). Studies show that interventions aimed to improve children's self-efficacy and coping strategies contribute significantly to their levels of motivation and learning, socio-cognitive functioning, emotional well-being and performance achievements (Katz, 2015; Usher et al., 2019). Thus, the importance of prevention, early diagnosis and treatment cannot be overstated, especially for African nations whose populations are among the youngest in the world.

The United Nations Convention on the Rights of the Child and scholars acknowledge the value of children's participation in research and helping efforts as instrumental in advancing their rights and interests alongside facilitating more effective interventions (Aldiss et al., 2009; Davies & Wright, 2008; Ruiz-Casares et al., 2013; United Nations General Assembly, 1989). Cognitive-behavioral-based mental health interventions focused on teaching social and emotional resilience skills directly to children and youth have been shown to be most effective in preventing and treating anxiety and depression (Ahlen et al., 2018; Coull & Morris, 2011; Mabrouk et al., 2022; Neil & Christensen, 2009). The FRIENDS Resilience program is a well-validated emotional resilience program utilized in cross-cultural settings worldwide. It is a cognitive-behavioral, emotional resilience intervention program that has been found to be effective in the prevention and treatment of anxiety and depression in children and adolescents. The program consists of four developmentally tailored modules: 'Fun FRIENDS' for children ages 4–7 (Barrett, 2007); 'FRIENDS for Life' for

children ages 8–11 (Barrett, 2010a); 'My FRIENDS Youth' for adolescents ages 12 to 15 (Barrett, 2010b, 2011a) and 'Adult Resilience' for ages 16 and above (Barrett, 2011b).

My FRIENDS Youth program teaches essential interpersonal development skills (such as building self-esteem, self-awareness, problem-solving and emotional regulation), as well as coping and management of anxiety and depression symptoms. This program has been implemented universally with large groups as well as smaller targeted groups and has shown to be effective among children across various cultural contexts (Fisak et al., 2023; Gallegos-Guajardo et al., 2015; Maalouf et al., 2020; Siu, 2007). Findings suggest that, compared to a control group, participants in the My FRIENDS program exhibited more significant improvements in depression and anxiety symptoms, proactive coping skills, psychosocial difficulties, self-concept, hope (Gallegos-Guajardo et al., 2015), self-esteem, internalizing symptoms (Siu, 2007), behavioral problems, specific and generalized anxiety symptoms and overall stress reduction (Sabey et al., 2019). However, despite its wide use, this evidence-based prevention program has not been used among sub-Saharan African children. The one quasi-experimental study to date assessed the 'FRIENDS for Life' program among South African children (Mostert & Loxton, 2008). Therefore, this pilot study explores the feasibility and effectiveness of the My FRIENDS Youth program in reducing mental health symptoms and improving relational outcomes among pre-adolescent and adolescent children in Zambia. Specifically, it was hypothesized that participants in the My FRIENDS Youth Program would demonstrate improvement in child-reported anxiety and depression symptoms, parent-reported internalizing and externalizing symptoms, attention problems and child-parent relationship quality.

## Method

### Participants

Participants consisted of 35 boys, 40 girls and their caregivers. The children were all fifth-grade students recruited from four government schools. Participants ranged in age from 10 to 15 years, with an average age of 12.1 (SD=1.02). Participants resided in urban low-income neighborhoods, referred to as 'Komboni' or 'compound' in the Zambian capital of Lusaka. Kombonis are typically informal housing or shanty towns commonly found in Zambian cities. These neighborhoods are characterized by high population density, low income, poor housing and sanitation, high crime and insufficient social amenities (Bwalya & Kabubi, 2019; Mutenje, 2019). The average household monthly income was 2,126.55 ZMW (SD=2,936.51). This is equivalent to $92 US dollars a month at an estimated $3 a day for average household sizes of 5.77 (SD=2.14) people in our sample. The 75 participants were comprised children (53% female) whose parents were 40 years old on average (SD=11.23), female (82%), married (62%) and had a primary (41%) or some secondary (39%) educational attainment. Descriptive statistics for participant demographics and study variables under the two conditions can be found in Table 1. Table 1 also includes analyses of group differences (intervention vs. control) on all variables. Baseline measures of control and outcome variables did not significantly differ across intervention or waitlist control groups, except for parent-child relationship outcomes, where the intervention group reported higher positive relationships and higher conflict.

**Table 1.** Baseline differences in study variables by group assignment (N = 75)

| Variable | Intervention (n = 44) | | Control (n = 31) | | Comparison | |
|---|---|---|---|---|---|---|
| | n | % | n | % | $\chi^2$ | p-Value |
| Youth gender | | | | | | |
| Male | 20 | 46.51 | 15 | 48.39 | 0.03 | .873 |
| Female | 23 | 53.49 | 16 | 51.61 | | |
| Parent gender | | | | | | |
| Male | 7 | 23.33 | 3 | 11.54 | 1.32 | .250 |
| Female | 23 | 76.67 | 23 | 88.46 | | |
| Parent marital status | | | | | | |
| Not married | 15 | 45.45 | 8 | 29.63 | 1.57 | .210 |
| Married | 18 | 54.55 | 19 | 70.37 | | |
| Parent education | | | | | | |
| Primary or less | 16 | 54.44 | 7 | 26.92 | 4.99 | .083 |
| Some secondary | 8 | 26.67 | 14 | 53.85 | | |
| Secondary+ | 6 | 20.00 | 5 | 19.23 | | |
| Caregiver relation | | | | | | |
| Parent | 30 | 73.17 | 22 | 73.33 | 2.36 | .308 |
| Grandparent | 8 | 19.51 | 3 | 10.00 | | |
| Aunt/Uncle/Sibling | 3 | 7.32 | 5 | 16.67 | | |
| | Mean | SD | Mean | SD | t | p-value |
| Household income | 1,510.97 | 1,778.55 | 2,833.33 | 3,779.42 | 1.74 | .087 |
| Parent age | 41.67 | 11.51 | 38.58 | 10.66 | −1.04 | .305 |
| Youth age | 12.17 | 1.17 | 11.97 | 0.80 | −0.82 | .416 |
| Anxiety | 4.03 | 3.56 | 4.73 | 3.73 | 0.81 | .422 |
| Depression | 4.83 | 3.36 | 6.40 | 3.95 | 1.81 | .075 |
| Internalizing symptoms | 3.28 | 2.45 | 3.57 | 2.01 | 0.50 | .621 |
| Externalizing symptoms | 4.19 | 3.46 | 3.57 | 3.41 | −0.69 | .492 |
| Attention problems | 3.56 | 2.46 | 4.21 | 3.02 | 0.92 | .361 |
| Positive parent-child relationship | 4.36 | 0.78 | 3.38 | 1.00 | −3.46 | .001 |
| Parent-child conflict | 1.75 | 0.61 | 2.40 | 1.09 | 2.32 | .026 |

## Procedure

Randomized cluster sampling was used to identify schools that would participate in the randomized controlled trial (RCT). A list of random locations in the Lusaka district in Zambia was generated using an online number generator, and high-income neighborhoods were removed using data from the Zambia Subnational poverty mapping report (De la Fuente et al., 2015). A list of public primary schools in these low-income neighborhoods was generated. From this list, four primary schools were randomly selected. Permissions for school participation in the RCT were obtained from the Zambia Ministry of Education and school administrators. Ethical approval was obtained from the University of Zambia and Brigham Young University.

Within each school, a universal recruitment approach was used. All fifth-grade classrooms (4 in each school) were approached, and students received caregiver information packets with a screening questionnaire. Completed packets were returned, collected and scored. Parents of children with scores greater than or equal to 15 on the Pediatric Symptom Checklist, 17-item (PSC-17) – indicating clinically significant levels of dysfunction – were invited to enroll their children in the study and to participate themselves. Children could participate with caregiver permission, even if caregivers opted out themselves. Informed consent from all participating parents and assent from youth were obtained. Randomization to intervention or waitlist-control groups took place after the completion of baseline assessments, with schools as the unit of randomization. First, schools were randomly ranked. Using Research Randomizer (randomizer.org), two sets of numbers corresponding to intervention or control conditions were randomly generated and applied to the list of ranked schools.

The final sample comprised 75 children (intervention n = 44, control n = 31). See Figure 1 for the flow of participants through each stage of the trial. Participants completed a researcher-administered baseline questionnaire assessing basic sociodemographic characteristics and key outcomes of interest. Questionnaires were administered

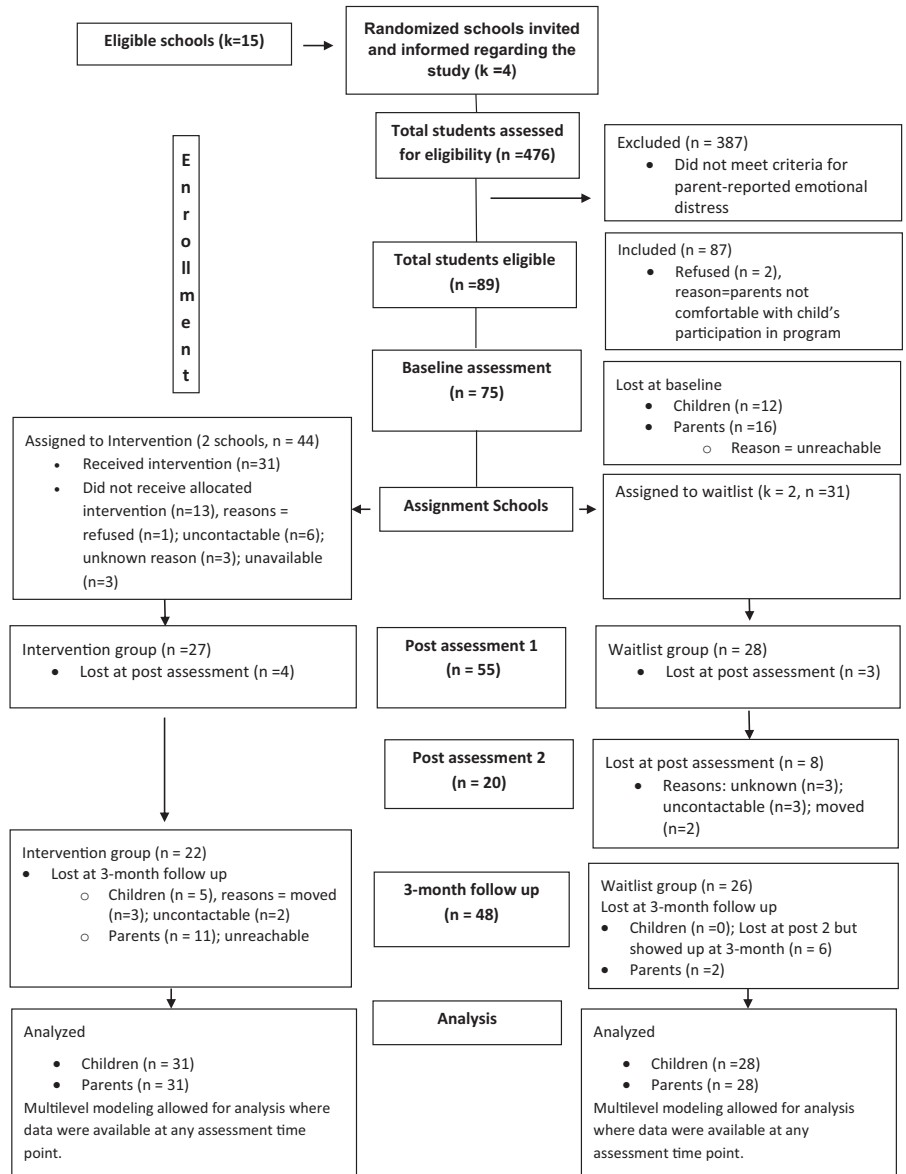

**Figure 1.** Consort flow diagram.

verbally on school grounds by local research assistants (blind to condition/cluster allocation), who entered data in Qualtrics. Youth in the intervention group participated in data collection across three time points: Baseline (T0), Post-Intervention (T1) and 3-Month Follow-Up (T3). Because the youth in the control group had two pre-tests prior to their intervention and post-follow-ups, they participated in data collection across four time points: Baseline (T0), Second Baseline (concurrent with the intervention group's Post-Intervention; T1), Post-Intervention (T2) and 3-Month Follow-Up (T3). Parents in both the intervention and control groups only participated in data collection at two time points: Baseline (T0) and 3-Month Follow-Up (T3).

## Intervention protocol

The My FRIENDS Youth Program was group-based and facilitated by a team of two to three mental health service professionals who received a 2-day training and certification from the FRIENDS resilience organization. The mental health service professionals were recruited from local mental health agencies, had a minimum of a bachelor's degree in psychology, and had experience working in the field of mental health. In conjunction with the 2-day certified program training, facilitators administering the intervention also participated in an additional 1-day training conducted by the first author, a licensed mental health provider. The training covered various aspects, including ethical considerations, group leadership skills and cultural adaptation of program language and content. Facilitators received weekly supervision to ensure program fidelity, provide feedback on intervention delivery, troubleshoot issues collaboratively and enhance overall program effectiveness. Teacher involvement with the program was minimal, limited to class announcements and providing academic records.

Group sessions were delivered as outlined in the My FRIENDS Youth Leader manuals (Barrett, 2010b) with a few modifications for cultural/environmental context as follows: (1) Owing to resource constraints in the schools and the limited time allocated to the after-

school program, sessions were condensed to 6 weeks midterm instead of the longer timeline (e.g., a school term or 12 weeks) typical in the original My FRIENDS Youth program. Group sessions (1.5 hours each) were held at a designated time twice weekly on school grounds for six consecutive weeks (12 sessions total). The sessions were delivered in groups of eight to twelve children per group; (2) Language was simplified, considering varying reading abilities and English comprehension, with facilitators assisting in writing tasks; (3) Stories and coping strategies were modified for cultural relevance, addressing locally identified challenges like hunger and homelessness with culturally appropriate coping methods. For example, coping strategies such as "taking a dog for a walk" were replaced with culturally relevant ways of coping, such as timely completion of chores to prevent disciplinary measures, such as spanking and how to demonstrate respect when interacting with elders and others to strengthen communal bonds valued in the culture for emotional support.

The content of the My FRIENDS Youth Program was delivered in the following fashion: **Session 1**: program introduction, reflect and define personal goals; **Session 2**: understanding feelings and empathy, reflecting on different ways of communication; **Session 3**: confidence building; **Session 4**: focusing on the present and becoming more aware, self-regulation; **Sessions 5 and 6**: attention training and self-talk, understanding influence of thoughts on feelings and behaviors, thought challenging; **Sessions 7 and 8**: coping strategies, problem-solving and building support teams; **Session 9**: managing interpersonal conflicts, bullying, self-care; **Session 10**: using the FRIENDS skills to help self and others; ***Booster Session 1***: review, ***Booster Session 2***: review and skill practice.

### Measures

*Anxiety and Depression.* Our primary outcomes, anxiety and depression were self-reported by youth and measured at all time points using the Generalized Anxiety Disorder-7 (GAD-7; Spitzer et al., 2006) and Patient Health Questionnaire-9 (PHQ-9; Kroenke et al., 2001), respectively. The GAD-7 and PHQ-9 items were measured on a 4-point scale (0=*not at all*, 1=*several days*, 2=*more than half the days*, 3=*nearly every day*). Items were summed (anxiety range: 0–15, depression range: 0–14). The internal reliability of both scales in our sample was good (anxiety: $\alpha = .83$, depression: $\alpha = .74$).

*Internalizing Symptoms, Externalizing Symptoms and Attention Problems.* Our secondary outcomes, youth internalizing symptoms, externalizing symptoms and attention problems were measured via parental report at the Baseline and 3-Month Follow-Up time points using the Pediatric Symptom Checklist (PSC; Jellinek et al., 1988). The PSC included 17 items on a 3-point scale (0=*never*, 1=*sometimes*, 2=*often*); five items measured internalizing symptoms, seven items measured externalizing symptoms and five items measured attention problems. The internal reliability for these subscales in our sample was sufficient (internalizing: $\alpha = .70$, externalizing: $\alpha = .79$, attention: $\alpha = .73$).

*Parent-Child Relationship.* Another secondary outcome, parent-child relationship outcomes was measured via parental report at the Baseline and 3-Month Follow-Up time points using the Child-Parent Relationship Scale (Pianta, 1992). The scale included 15 items measured on a 5-point scale (1=*definitely does not apply*, 2=*not really*, 3=*neutral or not sure*, 4=*applies somewhat* and 5=*definitely applies*); eight items measured relationship conflict and seven items measured positive relationship quality. The internal reliability of both subscales in our sample was sufficient (conflict: $\alpha = .78$, positive: $\alpha = .84$).

*Sociodemographic Controls.* Control variables included parent and child gender, which were dichotomous (0=*male*, 1=*female*); parent and child age, which were continuous and measured in years; caregiver relationship to child, which was categorical (*parent* [comparison]*, grandparent, aunt/uncle/sibling*); parent-reported household income, which was measured in Zambian Kwacha (ZMW; range: 0–20,000); parent marital status was dichotomous (0=*not married*, 1=*married*); parent educational attainment was categorical (*primary or less* [comparison]*, some secondary, secondary or higher*).

*Participants' Program Evaluation.* We assessed participants' subjective experiences with the program by using three questions. The first was a yes/no question that asked, "Do you feel that participating in the group was helpful for you?" We then asked the following open-ended questions: "Can you tell me more about things that you found helpful?" and, "Can you tell me more about the things that were unhelpful for you?"

### Analytic approach

Descriptive statistics and study analyses were conducted using Stata version 18 (StataCorp, 2023). Data were scanned for outliers and multicollinearity, neither of which were found. Missing data on the dependent variables was fairly minimal for anxiety and depression at T0 (<7%) but was higher at T3 (36%). Missing data on parent-reported measures was higher due to difficulty with parental involvement (T0 range: 20%–48%; T3: <26%). Intervention efficaciousness was examined using multilevel modeling, with time point (level 1) nested within person (level 2) and nested within schools (level 3). Models were tested using maximum likelihood estimation, which handles missing data by using all available data and has been shown to produce less biased estimates with more accurate standard errors compared to Listwise deletion (Enders, 2010). All models included sociodemographic controls.

The efficaciousness of the intervention on anxiety and depression in the intervention group compared to the waitlist control group was tested by introducing an interaction effect between waves (from T0 to T1) and group assignment (intervention vs. control). A statistically significant interaction effect would indicate that the intervention group experienced a change in Baseline and Post-Intervention scores was stronger than the change the control group experienced over the same period.

Because youth internalizing symptoms, externalizing symptoms, attention problems and parent-child relationship outcomes were only measured at two time points (with no intervention -control comparison), the multilevel models for these outcomes were quasi-experimental and examined the association between waves (from Baseline to 3-Month Follow-Up) and outcomes. A statistically significant beta coefficient would indicate that parents reported a change in outcomes between the Baseline and 3-Month Follow-Up (post-intervention) time points. We also included an interaction effect between waves (from T0 to T3) and group assignment (intervention vs. control) to determine whether these changes differed across groups. Thematic analysis was used to analyze the open-ended questions on children's experience with the intervention (Braun et al., 2022).

### Results

Means and standard deviations for the outcome variables of interest over time are shown in Table 2. Results from multilevel models

**Table 2.** Means and standard deviations of outcome variables over time (*N* = 75)

| | Intervention (*n* = 44) | | | Control (*n* = 31) | | | |
|---|---|---|---|---|---|---|---|
| Variable | T0 | T1 | T2 | T0 | T1 | T2 | T3 |
| Anxiety | 4.03 (3.56) | 4.74 (2.96) | 2.82 (2.26) | 4.73 (3.73) | 4.89 (3.38) | 3.95 (2.14) | 1.92 (2.08) |
| Depression | 4.83 (3.36) | 3.70 (2.09) | 4.00 (2.74) | 6.40 (3.95) | 4.93 (2.83) | 3.65 (2.01) | 2.34 (2.15) |
| Internalizing | 3.28 (2.45) | – | 1.77 (1.81) | 3.57 (2.01) | – | – | 0.96 (1.22) |
| Externalizing | 4.19 (3.46) | – | 0.97 (1.81) | 3.57 (3.41) | – | – | 1.35 (2.42) |
| Attention | 3.56 (2.46) | – | 1.77 (2.01) | 4.21 (3.02) | – | – | 1.42 (1.50) |
| PCR, Positive | 4.36 (0.78) | – | 4.85 (1.14) | 3.38 (1.00) | – | – | 4.69 (0.91) |
| PCR, Conflict | 1.75 (0.61) | – | 2.01 (1.05) | 2.40 (1.09) | – | – | 2.47 (1.30) |

*Note*: PCR, parent-child relationship. Standard deviations are in parentheses. T0, Baseline; T1 for Intervention group, Post-Intervention; T1 for control group, Second Baseline; T2 for intervention group, 3-Month Follow-Up; T2 for control group, Post- Intervention; T3 for intervention group, 3-Month Follow-Up. Dashes indicate that data for the variable was not collected at that time point.

where interactions were tested between intervention and control groups are shown in Table 3.

Results from multilevel models testing longitudinal changes in outcome variables for intervention and control groups are shown in Table 4. Results from the models predicting youth anxiety suggested that, compared to the waitlist control group, the intervention group did not show decreases in anxiety from Baseline to Post-Intervention (interaction $b = 0.55$, $p = .637$; see Table 3). However, the intervention group did show decreased anxiety from Post-Intervention to 3-Month Follow-Up ($b = -2.26$, $p < .01$). The control group did not show changes in anxiety from Second Baseline to Post-Intervention ($b = -1.09$, $p = .167$); however, the control group did show decreased anxiety from Post-Intervention to 3-Month Follow-Up ($b = -1.92$, $p < .001$).

Results from the models predicting youth depression indicate that, compared to the waitlist control group, the intervention group did not show decreases in depression from Baseline to Post-Intervention (interaction $b = 0.15$, $p = .877$; see Table 3). The intervention group did not show changes in depression from Post-Intervention to 3-Month Follow-Up ($b = 0.22$, $p = .767$; see Table 4). However, the waitlist control group did show decreased depression from Second Baseline to Post-Intervention ($b = -1.22$, $p = .046$) and further decreases from Post-Intervention to 3-Month Follow-Up ($b = -1.22$, $p = .014$; see Table 4).

The PSC (internalizing symptoms, externalizing symptoms, attention problems) and parent-child relationship multilevel models assessed changes from Baseline to 3-Month Follow-Up. Results from the PSC models show that, from Baseline to 3-Month Follow-Up, participants reported decreases in internalizing symptoms ($b = -0.32$, $p < .001$; see Table 3), externalizing symptoms ($b = -0.23$, $p < .001$; see Table 3) and attention problems ($b = -0.29$, $p < .001$; see Table 3). Interaction effects indicate that these

**Table 3.** Results from multilevel models testing interactions between wave and intervention assignment

| Independent variable | Anxiety | Depression | Internalizing symptoms | Externalizing symptoms | Attention problems | Parent-child relationship conflict | Positive parent-child relationship |
|---|---|---|---|---|---|---|---|
| Wave (Baseline, Post-Int) | 0.95 | −1.07* | – | – | – | – | – |
| Wave (Baseline, 3M F/U) | – | – | −0.32*** | −0.23*** | −0.29*** | 0.26 | 0.74*** |
| assignment | −0.18 | −1.10 | −0.02 | 0.04 | 0.05 | −0.49 | 0.56* |
| Wave×Assignment | 0.55 | 0.15 | 0.27 | −0.09 | 0.20 | −0.02 | −0.94* |
| Youth female | 0.71 | 0.51 | 0.02 | −0.17* | −0.14 | −0.31 | 0.07 |
| Youth age | 0.07 | 0.04 | 0.08 | −0.05 | −0.06 | −0.20 | 0.20 |
| Parent age | 0.02 | 0.04 | 0.00 | 0.00 | 0.00 | 0.00 | 0.00 |
| Parent female | 0.08 | 0.26 | −0.21* | 0.02 | 0.05 | 0.01 | −0.12 |
| Caregiver grandparent | 0.13 | −0.35 | 0.05 | 0.14 | 0.01 | 0.10 | −0.10 |
| Caregiver aunt/uncle/ sibling | −1.52 | −1.52 | −0.18 | −0.07 | −0.04 | 0.36 | 0.25 |
| HH income | 0.00 | 0.00 | 0.00 | 0.00 | 0.00 | 0.00 | 0.00 |
| caregiver marital status | 0.84 | 0.11 | −0.07 | −0.08 | 0.25** | 0.21 | 0.20 |
| Caregiver some secondary Ed | −0.22 | −0.35 | 0.04 | 0.03 | −0.09 | −0.04 | 0.23 |
| Caregiver secondary Ed+ | −0.45 | 0.21 | −0.05 | −0.07 | −0.22 | −0.43 | 0.43 |

*Note*: 3M F/U, 3-Month Follow-Up. The "Post-Int" measure was the control group's Second Baseline measure. All coefficients are unstandardized betas. Comparison category for caregiver relationship is "Caregiver Parent." Comparison category for caregiver education is "Caregiver Less Than High School." The analyses for all outcomes except youth anxiety and depression were quasi-experimental, as they were only measured at Baseline and 3-Months Post-Intervention. *$p < .05$, **$p < .01$, ***$p < .001$.

**Table 4.** Results from multilevel models testing changes in outcomes across time for intervention and waitlist control groups

| Outcome variable | Intervention group | | | Waitlist control group | | | |
| --- | --- | --- | --- | --- | --- | --- | --- |
| | Baseline – Post-Int | Post-Int – 3M F/U | Baseline – 3M F/U | Baseline – Second Baseline | Second Baseline – Post-Int | Post-Int – 3M F/U | Baseline – 3M F/U |
| Anxiety | 1.21 | −2.26** | −1.01 | 0.59 | −1.09 | −1.92*** | −2.45*** |
| Depression | −1.00 | 0.22 | −1.07 | −1.15 | −1.22* | −1.22* | −3.57*** |
| Internalizing symptoms | – | – | −1.16* | – | – | – | −2.40*** |
| Externalizing symptoms | – | – | −2.00*** | – | – | – | −1.24* |
| Attention problems | – | – | −0.22* | – | – | – | −0.42*** |
| Parent-Child relationship conflict | – | – | 0.42 | – | – | – | 0.31 |
| Positive parent-child relationship | – | – | 0.33 | – | – | – | 1.29*** |

*Note*: 3M F/U, 3-Month Follow-Up. All coefficients are unstandardized betas. All models included the sociodemographic controls shown in Table 3. All outcomes except youth anxiety and depression were only measured at the Baseline and 3-Month Follow-Up time points. *$p < .05$, **$p < .01$, ***$p < .001$.

reductions did not significantly differ based on group assignments (Internalizing interaction $b = 0.28$, $p = .054$; Externalizing interaction $b = −0.09$, $p = .474$; Attention interaction $b = 0.20$, $p = .193$; see Table 3).

Results from the parent-child relationship models show that, from Baseline to 3-Month Follow-Up, participants did not report a change in parent-child relationship conflict ($b = 0.26$, $p = .252$; see Table 3), and this did not differ based on group assignment (interaction $b = −0.02$, $p = .960$; see Table 3 and Figure 8 in the Supplementary Material) (Saasa et al. supplementary material 8). However, participants did report increases in the positive parent-child relationship ($b = 0.74$, $p < .001$; see Table 3), and this increase was stronger for participants assigned to the waitlist control group (interaction $b = −0.94$, $p = .019$; see Table 3).

### Program benefits

Results from the dichotomous question show that 94% of the children reported that the program was helpful for them. Qualitative data indicates that the children positively experienced the program. Themes from the children's responses show that the mental health program was helpful to them in building self-confidence (20%), teaching coping mechanisms and positive thinking (28%), problem-solving and goal setting (4%), handling bullying (13%), fostering positive social relationships (11%), learning relaxation techniques (9%), understanding inner thoughts and emotional regulation (9%). Example statements from child participants include: "*I learnt how to relax and how to solve problems and also how to set goals to achieve what I want to in my life*," "*I learned how to play well with my friends and not to fight*," "*I now have confidence in life and not being angry all the time*," "*I used to worry about a lot of things but not anymore.*"

Qualitative responses indicated that the program was well-rounded in supporting various aspects of children's mental and emotional well-being. Several children noted that providing snacks during sessions was valuable (9%). This may be an essential consideration in situations where food insecurity is high, as it not only addresses immediate physiological needs but also plays a critical role in creating an environment conducive to mental and emotional healing and education. In response to the question asking what was unhelpful with the program, the children either said nothing was unhelpful or that everything was helpful. It is possible that social desirability influenced these responses.

### Discussion

This study aimed to test the feasibility and efficacy of the My FRIENDS Youth Program, a universal cognitive-behavioral intervention for anxiety prevention and resilience enhancement among Zambian youth. To our knowledge, this is the first randomized controlled trial of this program in the Zambian context. Access to mental health services in sub-Saharan Africa is extremely low despite the disproportionately high levels of mental health issues among children in this region (Jörns-Presentati et al., 2021). Therefore, providing and testing culturally informed mental health interventions in these locations is highly important, as is testing interventions that will reduce the burden placed on the already strained mental health service system (Chibanda et al., 2020). The My FRIENDS Youth Program has the potential to be both a highly effective and highly efficient program for Zambian youth, given the fact that it can be conducted in large group settings and involves minimal community and parental involvement. To further reduce the burden on parents and caregivers, we only gathered parent-reported measures at two time points (baseline and post-intervention), which made the analyses of those outcomes quasi-experimental. The RCT was conducted with a waitlist control group design, where the control group waited an additional 6 weeks to receive their intervention to serve as the comparison group for the intervention group's post-intervention scores. The intervention was administered by trained mental health professionals in line with studies that have shown clinician-administered programs to be most effective compared to non-clinician-administered (Fisak et al., 2023).

When examining pre- to post-intervention anxiety and depressive symptoms, the intervention group did not experience statistically significant decreases compared to the control group. However, both the intervention and control groups exhibited lower anxiety symptoms from Post-Intervention to 3-Month Follow-Up, and the control group showed lower depression symptoms from Second Baseline to Post-Intervention and then further decreases from Post-Intervention to 3-Month Follow-Up. This suggests a potential delayed effect of intervention benefits. These results are similar to other studies from both western and non-western

contexts where immediate reductions in anxiety symptoms were not observed until a longer time had passed (Barrett et al., 2006; Essau et al., 2012; Lowry-Webster et al., 2003; Mostert & Loxton, 2008; Ruttledge et al., 2016). In a sample of South African children (Mostert & Loxton, 2008, N=46), no significant differences in anxiety levels were found immediately post-intervention between the intervention group and control group, with symptom reductions manifesting later at 4- and 6-month follow-ups. Studies with much larger samples have also shown similar results. For instance, Essau et al. (2012) studied 634 German children across 14 schools, finding that 11–12-year-olds in the intervention group did not show immediate gains compared to controls but exhibited reduced anxiety and depression levels at 6- and 12-months post-intervention. Similarly, Ruttledge et al. (2016) observed no significant differences in anxiety reduction among 709 Irish children from 27 schools until several months after the intervention. These findings may arise for several reasons; participants may need more time to practice and master the skills for the intervention to be effective. Notably, the MY FRIENDS Youth Program was originally designed as a 12-week intervention (Barrett, 2010b), suggesting that our study's shorter duration may have been insufficient to capture immediate gains. Further, because this is a prevention program, the intervention may not only reduce symptoms but also prevent their expected increase over time (Michelson et al., 2020). A longer waitlist period in future studies could help track symptom changes without intervention. Additionally, since both groups started with low, non-clinical anxiety and depression levels, the intervention's effects may have been harder to detect. Future research could focus on participants with clinically significant symptoms for a clearer assessment.

We also examined secondary outcomes to ascertain additional insight into the interventions' broader effects. Parent-reported changes from Pre-Intervention to 3-Month Follow-Up showed no changes in parent-child conflict. It did show decreases in youth internalizing symptoms, externalizing symptoms, attention problems and increases in positive parent-child relationships. Similarly, Anticich et al. (2013) observed improvements in parent-child interactions, while reductions in externalizing behaviors were also noted following FRIENDS intervention (Anticich et al., 2013; Kozina, 2018). These findings suggest that the MY FRIENDS Youth Program may not only improve youth mental health symptomology but also improve parent-child relationships. These findings are consistent with other studies that show youth mental health interventions to be beneficial for broader family functioning, even when the family unit is not the focus of the intervention (Pedersen et al., 2019). However, given the strong link between parent and caregiver behaviors and youth mental health, caregiver engagement in youth interventions has been shown to facilitate better mental health outcomes (Barnett et al., 2020). Future research will need to replicate these findings and ensure that an intervention group is compared to a waitlist control group to confirm that these improvements are due to the intervention program and not other extraneous factors.

### Strengths and implications

This study's strengths include the random assignment of schools to intervention, the inclusion of a control group for child-reported measures, caregivers as additional informants, utilization of an evidence-based intervention and use of outcome measures that are highly reliable and validated across various cultures. This study also adds to the global discourse on the feasibility and effectiveness of the FRIENDS resilience program among sub-Saharan African children in schools.

The implications of implementing a Western-based mental health intervention among children in sub-Saharan Africa are multifaceted. First, the observed improvements from the 3-month post-intervention indicate that such interventions hold promise for addressing mental health challenges in this population. However, the lack of immediate effects underscores the need for sustained engagement and adaptation to local contexts. Additionally, the program was well-received, with noted gains reported by the children. In one school, children in higher level grades who were not participating in the program made an official petition with the school administration requesting access to similar services, thus indicating a desire for mental health support services among this population.

Despite the positive outcomes, several challenges need to be addressed: (1) The costliness of the intervention poses a significant barrier, highlighting the necessity for cost-effective alternatives tailored to resource-constrained settings in Africa; (2) Facilitator feedback regarding culturally irrelevant examples in the manual/workbook, participant difficulties with some English terms, and literacy-related issues for homework tasks point to the importance of cultural adaptation and linguistic accessibility. Moving forward, efforts should focus on refining the intervention to better align with the target population's cultural norms and linguistic capabilities. This may involve revising materials, incorporating local languages and providing additional support for children with literacy challenges. Moreover, the positive experiences reported by children participating in the program highlight the importance of continued efforts to tailor interventions to the specific needs and preferences of the target population. Collaborative partnerships between Western and African mental health professionals can facilitate the development of culturally sensitive interventions that maximize effectiveness and accessibility. Ultimately, addressing these challenges is crucial to ensure the successful implementation and sustainability of mental health interventions tailored to the needs of children in sub-Saharan Africa.

### Limitations

Study results should be considered in the context of study limitations. First, while the impact of the intervention on youth-reported measures (anxiety and depression) was analyzed using standard experimental design techniques for treatment-control comparisons, we could not do the same for parent-reported measures, which were measured only at Baseline and Post-Intervention. This was due to parental time constraints and contextual challenges in caregivers' participation. Therefore, the parent-reported models were quasi-experimental and did not involve a true control-group comparison. Thus, it is difficult to confirm whether the difference between pre-test and 3-month post-test scores on parent-reported measures are related to the intervention or participant maturation and passage of time. Second, statistical power and the external validity of the results might be constrained by the small number of clusters and sample size (Faber & Fonseca, 2014). Future studies with larger sample sizes, a predetermined number of clusters and cluster sizes, and comparison groups across all data collection time points would allow for increased detection of true effects in a more diverse population. However, despite our study's small sample size and resource-constrained setting, our findings align with those from larger

studies in high-income countries, reinforcing the intervention's broader applicability.

Third, as with most voluntary research, our sample was subject to self-selection bias. Further, outcome measures were solely based on parent and child-reported measures, which can be prone to bias. Comprehension of self-reported measures of depression and anxiety symptoms may have been impacted by the children's emotional and cognitive development, posing a limitation. Additionally, evidence shows that parent and child reports on psychopathology tend to deviate from each other (Caqueo-Urízar et al., 2022; Kuitunen-Paul et al., 2023). Thus, for more accurate measures of children's mental health symptoms, researchers can include pre- and post-assessments from mental health professionals. Scholars show that a combination of parent, self-reported and clinician-reported symptoms yield a more complete assessment of children's psychopathology (Cuijpers et al., 2010; Kuitunen-Paul et al., 2023; Youngstrom et al., 2003). Lastly, while program fidelity and quality control measures such as weekly monitoring and supervision were in place, factors such as time allotted to the intervention, facilitator characteristics and group dynamics may have influenced participants' responses to the intervention.

## Conclusion

Despite these limitations, the My FRIENDS Youth intervention seems promising in altering Zambian children's risk of developing psychopathology and addressing the mental health needs of this population. Future research may need to adapt this intervention further to meet the needs of children in Zambia better, considering the social, cultural and environmental contexts within which they function. In alignment with the global effort for scaling up mental health prevention efforts in low-resource settings, this study shows that school-based efforts that focus on psychological distress prevention and resilience enhancement are feasible and could make a significant impact on African children's social and emotional well-being.

**Open peer review.** To view the open peer review materials for this article, please visit http://doi.org/10.1017/gmh.2025.33.

**Supplementary material.** The supplementary material for this article can be found at http://doi.org/10.1017/gmh.2025.33.

**Data availability statement.** The datasets generated and/or analyzed during the current study are not publicly available due to ethical restrictions but are available from the corresponding author upon reasonable request.

**Acknowledgments.** The authors extend their heartfelt gratitude to the teachers, parents and children whose participation enriched this study. Special appreciation is also extended to the mental health team for facilitating the groups. We also wish to acknowledge all the students who participated in the study and subsequent follow-ups and the school coordinators for their invaluable administrative support.

**Author contribution.** S.S.: writing and conceptualization; K.P.W.: writing and data analysis; C.G.S.: writing and review; P.B.: editing and review; C.Y.L.: editing and review.

**Financial support.** This study was funded by the College of Family, Home and Social Sciences at Brigham Young University.

**Competing interests.** P.B. is the developer of the FRIENDS Resilience program. C.Y.L. is employed at FRIENDS Resilience organization. The remaining authors declare none.

**Ethics statement.** This study was conducted in accordance with the ethical standards of the University of Zambia, Brigham Young University, and with the 1964 Helsinki declaration and its later amendments or comparable ethical standards. Informed consent and assent, as appropriate, were obtained from all individual participants included in the study.

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
