## [Reviewer Report]

This paper reports what is effectively a cluster randomised controlled trial rather than a basic RCT, as the unit of randomisation is the school. There is no power calculation presented in the methods to predetermine number of clusters and cluster size. There is no differentiation of primary and secondary outcomes. There are only four schools in total, ie two schools in each arm of the study, which I suspect is too small a number of clusters and should probably be at least four schools per arm and possibly more. I do not know if the number of children recruited in each cluster is adequate for only two schools in each cluster or indeed if there had been at least four schools in each cluster. The flow diagram does not address how many children were in each fifth year class. It seems that children and parents were asked to express interest in the study ie to volunteer rather than being systematically approached and asked for consent. We do not know what proportion of children in each class expressed interest in the study, prior to being assessed for eligibility for inclusion. It therefore appears that the method of recruitment of children into each cluster was not systematic but rather a volunteer sample which would have led to selection biases. The numbers in the flow diagram do not always add up, eg out of 44 children assigned to treatment, 31 received it, and 10 did not, but what happened to the other 3??

It is important to know whether the research assistants were blind to cluster allocation?

The title of the paper needs to identify that this is a cluster RCT, and to indicate if it was blinded or double blinded or neither.

The limitations section of the paper does not address these issues.

I think the paper could conceivably be reported as a pilot study to assess feasibility of such a study in Zambian schools, to tighten the methodological procedures and to explore the need for more adaptation of the treatment method.

---

## [Reviewer Report]

I have read with interest the paper.

It adds to the literature regarding research to improve mental health of school-going children in low-income countries. However, I have a few revisions to suggest.

1.The FRIENDS program is essentially a prevention program, as such the authors would do well by emphasizing this from the outset and throughout the manuscript. There is some confusion regarding the use of the term intervention versus treatment, I would suggest they avoid the term treatment, despite the trial suggesting treatment effects were measured. This way, the lack of significant improvement or unexpected results across both case and control groups would be understandable.

2.Please consider indicating you obtained assent and not consent from the child participants.

3.Consider discussing results in context and compare with trials in low-income countries if any, in part because the challenges of doing research in a relatively unstructured school environment might be unique.

4.It is unclear why the interviews, despite the challenges of comprehending the questionnaires, were self-adminstered by the students,and not interviewer-administered. Could this have affected completion rates towards the end of the study?

5.Please describe more clearly the Time points for the control group, Do they complete questionnaires at additional time points after the intervention groups have concluded? explain the extra time point for the control group.

6.To what extent were the class teachers involved and were specific times allocated during the two meetings per week over 6 weeks? Were the groups meetings at the beginning, middle or towards the end of the school term?

7.The results based on the GAD and PHQ-9 scores across the two groups are not reflected. It is possible that the lack of improvement or change in anxiety or depression symptoms might be due to the fact that across both groups, levels of anxiety and depression were not clinically significant in the first place.

8.It is unclear if there were any untoward effects of the study among participants, given the potential distress among those included in the study based on having higher scores on GAD and PHQ-9. Did any of the participants need counseling or otherwise?

Thank you.

---

## [Editor Report]

Dear Authors,

Your manuscript: “Efficacy of a School-Based Mental Health Intervention Among Zambian Youth: A Randomized Controlled Trial”, has now been reviewed,

---

## [Editor Report]

Dear Ms Saasa,

Your revised manuscript titled “Efficacy of a School-Based Mental Health Intervention Among Zambian Youth: A Cluster-Randomized Controlled Trial” has now been reviewed

A minor comment on page 5, line 3 reading..’Thus, the importance of prevention, early diagnosis and treatment cannot be overstated..‘ In public health, it is understood that early diagnosis and treatment are components of prevention(Secondary prevention). I suggest that the word ’primary' be inserted before prevention.